# Bayesian probabilistic projections of proportions with limited data: An application to subnational contraceptive method supply shares

Hannah Comiskey[1]*, Niamh Cahill[2], Leontine Alkema[3], David T. Frazier[1], Worapree Maneesoonthorn[1]

1 Department of Econometrics and Business Statistics, Monash University, Melbourne, Australia, 2 Department of Mathematics and Statistics, Maynooth University, Kildare, Ireland, 3 Department of Biostatistics and Epidemiology, University of Massachusetts Amherst, Amherst, Massachusetts, United States of America

* hannah.comiskey@monash.edu

## Abstract

Engaging the private sector in contraceptive method supply is critical for creating equitable, sustainable, and accessible healthcare systems. To achieve this, it is essential to understand where women obtain their modern contraceptives. While national-level estimates provide valuable insights into overall trends in contraceptive supply, they often obscure variation within and across subnational regions. Addressing localised needs has become increasingly important as countries adopt decentralised models for family planning services. Decentralization has also underscored the need for reliable subnational estimates of key family planning indicators. The absence of regularly collected subnational data has hindered effective monitoring and decision-making. To bridge this gap, we propose a novel approach that leverages latent attributes in Demographic and Health Survey (DHS) data to produce Bayesian probabilistic projections of contraceptive method supply shares (the proportions of modern contraceptive methods supplied by public and private sectors) with limited data. Our modelling framework is built on Bayesian hierarchical models. Using penalized splines to track public and private supply shares over time, we leverage the spatial nature of the data and incorporate a correlation structure between recent supply share observations at national and subnational levels. This framework contributes to the domain of subnational estimation of proportions in data-sparse settings, outperforming comparable and previous approaches. As decentralization continues to reshape family planning services, producing reliable subnational estimates of key indicators is increasingly vital for researchers and policymakers.

**Data availability statement:** All files are available from: https://github.com/hannahcomiskey/Comiskey_PlosOnepaper.

**Funding:** Worapree Maneesoonthorn and Hannah Comiskey gratefully acknowledge support by the Australian Research Council through grant DP200101414. David Frazier gratefully acknowledges funding from the Australian Research Council under Projects DE200101070 and DP200101414. The funders had no role in study design, data collection and analysis, decision to publish, or preparation of the manuscript.

**Competing interests:** The authors have declared that no competing interests exist.

## Introduction

FP2030 is a global partnership committed to empowering women and girls through investments in rights-based family planning [1]. This approach emphasizes accessibility and availability of contraceptives [2]. Traditionally, the public sector has led family planning efforts [3], but declining donor funding has spurred interest in private sector engagement [4–6]. Leveraging private sector contributions enhances equitable, accessible, and sustainable reproductive healthcare [7,8]. Recent initiatives, such as WHO's 2024 online training for pharmacists, aim to expand contraceptive access in low- and middle-income countries [9]. Pharmacies play a crucial role, particularly in providing short-term methods like oral contraceptives and condoms [10]. Unmarried women and teenagers disproportionately rely on the private sector due to privacy concerns [11]. Therefore, understanding contraceptive access patterns is essential for equity [8,11]. National estimates of family planning indicators aid global reporting but overlook local trends and fail to assess if target levels for indicators are being ubiquitously attained across the nation. Furthermore, a decentralised approach to family planning management improves sector performance [12,13]. Subnational estimates are widely used in demography for tracking contraceptive prevalence, unmet need, and under-5 mortality [14–16], yet gaps remain in monitoring contraceptive supply sources.

Small area estimation (SAE) involves estimating characteristics of sub-populations, where there may be few or no samples available, by 'borrowing strength' from related populations to improve accuracy [17]. Using SAE approaches to model demographic and health outcomes is a well-established practice. In 1979, the foundational Fay-Herriot modelling approach described a two-stage hierarchical area-level model that allowed for the weighted estimates of area-level characteristics [18]. Building on this, in 1991 Basag, York, and Mollie (BYM) described an approach that uses both spatially structured variation and independent unstructured variation to capture underlying disease prevalence [19]. Since these models, the area of SAE has exploded with progress in many different directions including generalised linear mixed models, Bayesian hierarchical models, and conditional auto-regressive (CAR) models [20–24].

Previous works modelled the distribution of contraceptive method supply shares at the national level using a Bayesian hierarchical penalized spline model, and then explored the application of this modelling framework to subnational data [25,26]. While performing relatively well across a range of validation measures, this paper revisits the modelling framework of Comiskey et al. (2024) with the aim of improving its suitability for estimating and projecting proportions, while addressing the complex structure of subnational DHS administrative data. In this article, we describe a Bayesian hierarchical penalized spline model that produces annual, subnational, and method-specific estimates of the proportion of modern contraceptives coming from the public sector. In the context of SAE, Bayesian hierarchical models allow for the pooling of information across larger populations to inform smaller sub-population estimates, where data is sparser. As a result, these models provide a powerful framework for estimating subnational method supply shares. The Comiskey et al. (2024)

approach also estimates and incorporates cross-method correlations to inform the rates of change in spline coefficients. However, due to the heterogeneity of subnational data, capturing cross-method correlations within the rates of change between spline coefficients becomes difficult. Instead, we present an approach that captures the cross-method correlations for the most recently observed survey levels of public sector supply shares. Finally, Comiskey et al. (2024) estimate a multivariate compositional outcome across three-way breakdown of the contraceptive supply market. Due to small sample sizes at the subnational level, we instead focus on a simplified public–private breakdown of the outcome.

Overall, we have modified the national-level modelling approach of Comiskey et al. (2024) to subnational regions where the sampling density is lower (i.e., fewer women sampled per estimate) and consequently the uncertainty associated with the survey observations is higher than what is observed as the national level. Like Comiskey et al. (2024), the updated approach relies on Bayesian hierarchical estimation for key parameters, but introduces an updated prior structure within the proposed modelling framework. The proposed framework now provides estimates of the latent cross-method correlations that exist between the most recently observed levels of contraceptive method supply shares. Finally, the flexibility of this updated modelling approach captures the nature of an otherwise complex time- and spatially-varying dataset. At the time of writing, these subnational estimates with associated uncertainty represent a significant advancement in monitoring contraceptive supply shares. While our focus is the specific supply-share monitoring scenario, we note that this more parsimonious modeling approach is highly generalisable and can be applied to a wide range of datasets beyond the current context. By structuring the model to capture both hierarchical dependencies and flexible distributional assumptions, it can accommodate different levels of aggregation, varying sample sizes, and diverse outcome types; e.g., the proportion of women who received antenatal care from a skilled provider or proportion of children vaccinated between 12–23 months [27–30].

The outline of this paper is as follows. We begin with the definitions used throughout this paper and the data utilised in this approach. In the Methods section we describe the hierarchical penalised spline model. In Results, we consider the output of this model, which we then compare with various approaches listed in the Supplementary Materials, via an extensive model comparison study. We conclude the paper with a discussion. The Supplementary Materials contain more technical details and additional supporting information.

## Definitions and data sources

### Definitions

Modern contraceptive methods are defined as "a product or medical procedure that interferes with reproduction from acts of sexual intercourse" [31]. During this investigation, we consider five main modern methods of contraception. These are female sterilization, oral contraceptive pills (OC pills), implants (including Implanon, Jadelle and Sino-implant), intra-uterine devices (IUD, including Copper- T 380-A IUD and LNGIUS), and injectables (including Depo Pro-vera (DMPA), Noristerat (NET-En), Lunelle, Sayana Press and other injectables). Contraceptive method supply shares are defined as "the percent distribution of the types of service-delivery points cited by users as the source of their current contraceptive method (if more than one source, then the most recent one)" [32]. This paper considers a public/private sector breakdown for each of the five contraceptive methods listed. Contraceptives supplied by government-funded health facilities and home/community deliveries are considered to be supplied by the public sector, while supplies that come from sources outside the public sector, including enterprises ran for-profit or non-profit, can be defined as coming from the private sector.

### Data sources

Following the data curation approach described in Comiskey et al., we use a database of administration-1 level Demographic and Health Survey (DHS) observations for public and private sector modern contraceptive method supply shares with their associated standard errors [25]. This database was created on 01/09/2024 using the Integrated Public Use

Microdata Series (IPUMS)-DHS data [33]. The authors did not have access to information that could identify individual participants during or after data collection. The variables contained within the IPUMS-DHS database are consistent over time and space, making them suitable for temporal and spatial analysis. The countries included in this study are selected from those participating in the FP2030 initiative with IPUMS-DHS data available after 2012 and were also previously considered in Comiskey et al. 2024. There are 23 countries containing 160 subnational regions, included in this study. The administration level used was the integrated geographic units calculated by IPUMS. Using the integrated geographic units ensured that the data had consistent boundaries across all years. To avoid issues with exact zeros or ones, on the logit scale, the data was condensed using the 'lemon-squeezer' method [34]. This simple transformation slightly shrinks all data points away from 0 and 1 while maintaining their relative ordering and approximate scale. To address issues with small sample size, we filtered the database to only include observations where at least one sector (public or private) has a sample size of at least 10 women. This removes sets of observations with large uncertainty. Sampling errors were calculated while accounting for the sampling design using a Taylor series linearisation method to approximate the standard error of the calculated proportions [35,36]. A complete description of this imputation process is described in the Supplementary Materials.

Table 1 gives a summary of each country included in the dataset. In this dataset, Burkina Faso has the oldest 'latest' survey, collected in 2010. In contrast, Madagascar has the newest 'latest' survey collected in 2021. Recorded in this dataset, 56% of countries collected their latest surveys between 2015 and 2018, while a further 28% of countries collected

**Table 1. Summary information regarding the countries considered for subnational modelling. The name, number of subnational administration level 1 (admin-1) regions, the year of the first survey collected, the year of the most recently collected DHS survey, and the total number of DHS surveys are given for each country in the dataset.**

| Country | Total subnational regions | Earliest survey | Latest survey | Total surveys |
|---|---|---|---|---|
| Benin | 6 | 2001 | 2017 | 4 |
| Burkina Faso | 14 | 1993 | 2010 | 4 |
| Cameroon | 3 | 1991 | 2018 | 4 |
| Cote d'Ivoire | 15 | 1994 | 2011 | 3 |
| Ethiopia | 10 | 2000 | 2019 | 5 |
| Ghana | 8 | 1993 | 2014 | 5 |
| Guinea | 3 | 1999 | 2018 | 4 |
| Kenya | 7 | 1993 | 2014 | 5 |
| Liberia | 5 | 2007 | 2019 | 4 |
| Madagascar | 6 | 1992 | 2021 | 5 |
| Malawi | 3 | 1992 | 2016 | 5 |
| Mali | 4 | 1995 | 2018 | 5 |
| Mozambique | 11 | 1997 | 2011 | 3 |
| Myanmar | 15 | 2015 | 2015 | 1 |
| Nepal | 5 | 1996 | 2016 | 5 |
| Niger | 6 | 1992 | 2012 | 4 |
| Nigeria | 7 | 1990 | 2018 | 6 |
| Pakistan | 6 | 1991 | 2017 | 4 |
| Rwanda | 5 | 1992 | 2019 | 7 |
| Senegal | 4 | 1992 | 2017 | 9 |
| Tanzania | 6 | 1991 | 2015 | 6 |
| Uganda | 4 | 1995 | 2016 | 4 |
| Zimbabwe | 10 | 1994 | 2015 | 5 |

their latest survey in 2014 or before. Malawi, Guinea and Cameroon are divided into 3 subnational areas, making them the countries with the least number of subnational regions. The amount of survey data available also varies per country. Senegal has the largest number of surveys in the dataset with 9 surveys collected between 1990 and 2021, while Myanmar has only one survey present in the dataset.

## Methods

### Modelling approach

The outcome of interest is the components of a compositional vector,

$$\phi_{\boldsymbol{p,t,m}} = (\phi_{p,t,m,1}, \phi_{p,t,m,2}) \tag{1}$$

where $\phi_{p,t,m,s}$ is the proportion supplied by the public (s = 1) or the private (s = 2) sector of modern contraceptive method $m$, at time $t$, in subnational administration region $p$ and $\sum_{s=1}^{2} \phi_{p,t,m,s} = 1$ .

The documentation of the modelling assumptions is described using the standarised Temporal Models for Multiple Populations (TMMPs) framework [37]. First, we describe the process model which captures the latent dynamics of the outcome of interest. Secondly, we discuss the data model which links the observed data to the process model. In the process model, we model the logit-transformed proportion of the public sector supply share using a Bayesian hierarchical penalised spline model. The logit-transformed data is linked to the process in the data model via a Normal distribution.

Assuming a hierarchical estimation approach accounts for the spatial structure of the data, enabling subnational-level intercepts to benefit from information sharing within each country. Extending this assumption to a multivariate framework across methods allows the model estimates to further leverage latent cross-method correlations embedded in the covariance structure. A key advantage of this approach is the collective information sharing that occurs both across methods and within countries, spanning subnational administrative regions. This mitigates the effects of data sparsity observed in certain subnational regions and countries. The use of a random walk model of order 1 to estimate the coefficients of the penalised splines allows us to flexibly model non-linear trends, while the penalty term controls the degree of smoothness and prevents overfitting. Moreover, the approach builds on previous work that forecasts a steady state beyond the most recent survey observation [25,38].

**The process model.** The logit-transformed public sector proportion, logit($\phi_{p,t,m,1}$ ), is modelled via a hierarchical structure depicted in Fig 1. We construct the spline using,

$$\text{logit}\left(\phi_{p,t,m,1}\right) = \psi_{p,t,m} = \sum_{k=1}^{K} \beta_{p,m,k} B_{p,k}(t) \tag{2}$$

where, $\psi_{p,t,m}$ is the latent variable capturing the logit-transformed public sector proportion. $B_{p,k}(t)$ is the k$^{th}$ basis function evaluated at time $t$, in subnational administration region $p$. K is the total number of knots chosen for the set of basis functions. $\beta_{p,m,k}$ is the k$^{th}$ spline coefficient for method $m$, in subnational administration region $p$. Within basis functions, knots are the points where the piece-wise polynomials join together over the input variable (e.g., x-axis). Following the approach outlined in Comiskey et al. (2024), we align a knot with the year of the most recently observed survey in subnational administration region $p$. This results in each subnational administration region $p$ having a unique set of basis functions.

Aligning the knots of the basis function with the most recently observed survey year serves two purposes. Firstly, it allows steady state projections beyond the most recent survey year and secondly, it allows for a random walk model of order 1 estimation approach to the vector of K spline coefficients, $\boldsymbol{\beta_{p,m}}$ . Let k$^*$ be the knot that aligns with the year of the most recently observed survey, then

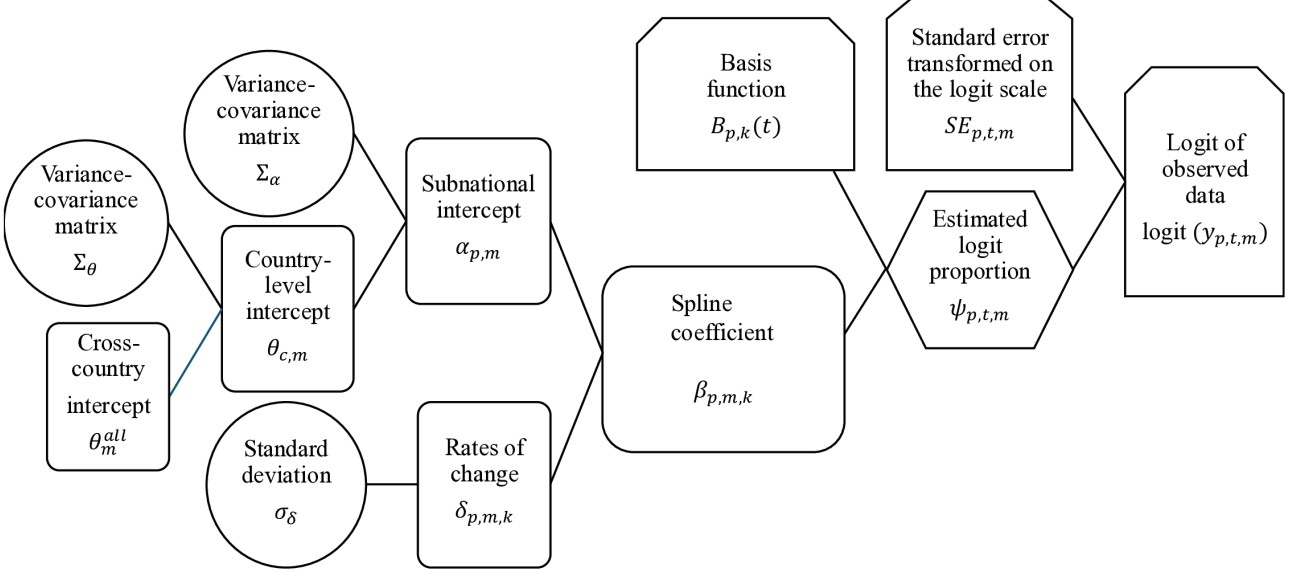

**Fig 1. Flow diagram of the subnational supply share model.** A flow diagram illustrating the relationships among the parameters contributing to the estimation of subnational supply shares. From left to right, circular nodes represent variance parameters that inform the second-level model components. Intercepts and rates of change are combined to generate spline coefficients (shown as rounded rectangles). These spline coefficients are then combined with subnational area–specific basis functions to produce the estimated proportions (shown as hexagons). The estimated proportions are linked to their associated standard errors, calculated from DHS survey data and transformed to the logit scale using the Delta method. Together, these quantities connect to the observed data (shown as clipped rectangles).

$$\beta_{p,m,k} = \begin{cases} \alpha_{p,m} & k = k^*, \\ \beta_{p,m,k+1} - \delta_{p,m,k} & k < k^*, \\ \beta_{p,m,k-1} + \delta_{p,m,k-1} & k > k^*. \end{cases}$$

(3)

where,

$$\delta_{p,m} = (\beta_{p,m,2} - \beta_{p,m,1}, \beta_{p,m,3} - \beta_{p,m,2}, \ldots, \beta_{p,m,K} - \beta_{p,m,K-1}).$$

(4)

We assume that the value of the spline coefficient at the point $k^*$ is $\alpha_{p,m}$. Moving backwards from $k^*$ to $k = 1$, we subtract the corresponding rates of change $\delta_{p,m,k}$. Similarly, moving from $k^*$ towards $k = K$, we add the corresponding rates of change $\delta_{p,m,k-1}$. We set a sum-to-zero constraint on the spline coefficients to ensure identifiability of the parameter estimates. The spline coefficient $\alpha_{p,m}$ acts as a proxy intercept in the modelling approach, as a change in $\alpha_{p,m}$ systemically shifts the set of public sector share projections. We specify a hierarchical prior using a multivariate Normal distribution, which is designed to capture both geographic and cross-method trends in the data. In this framework, subnational and method-specific trends are informed by country-level averages so that information is shared across regions and methods, which is known to improve inference in data-sparse settings, inducing a form of spatial dependence. We assume,

$$\theta_m^{all} \sim N(0, 100),$$

(5)

$$\theta_{c,1:M} \mid \theta_m^{all}, \Sigma_\theta \sim MVN(\theta_{1:M}^{all}, \Sigma_\theta),$$

(6)

$$\alpha_{p,1:M} \mid \theta_{c,1:M}, \Sigma_\alpha \sim MVN(\theta_{c,1:M}, \Sigma_\alpha),$$

(7)

where M is the total number of contraceptive methods considered. For country $c$ and set of M methods, we assume that the intercepts $\theta_{c,1:M}$ are centered on a cross-country method-specific mean $\theta_m^{all}$ with an associated variance-covariance matrix $\Sigma_\theta$. $\theta_{c,1:M}$ capture the most recently observed publicly supplied proportions at the national level and on the logit scale. $\Sigma_\theta$ is MxM matrix which is constant across all countries. Similarly, the M intercepts of subnational administration region $p$, $\alpha_{p,1:M}$, are centered on a set of country-specific means, $\theta_{c,1:M}$, with an associated cross-method variance-covariance matrix $\Sigma_\alpha$ that is constant across all subnational administration regions. Finally, the variance-covariance matrices assume vague inverse-Wishart priors, centered on an identity matrix of size MxM with M+1 degrees of freedom. The use of this prior assumes that the marginal distributions of the correlations between methods follow a uniform distribution. It is also conjugate prior to the multivariate Normal distribution, making it a suitable choice in our hierarchical estimation process [39].

$$\Sigma_\theta \sim IW(I^M, M+1), \tag{8}$$

$$\Sigma_\alpha \sim IW(I^M, M+1). \tag{9}$$

The rates of change between spline coefficients, $\delta_{p,m,k}$, are modelled assuming a Normal prior with cross-method variance. We penalise the spline coefficients to remain steady over time, by assuming the expected rate of change to be 0. This acts as a penalty for the B-splines. A vague truncated-Normal prior was used to capture this variation.

$$\delta_{p,m,k} \mid \sigma_\delta \sim N(0, \sigma_\delta) \tag{10}$$

$$\sigma_\delta \sim N(0, 2^2)_+. \tag{11}$$

**The data model.** We link the latent variable $\psi_{p,t,m}$ to the logit-transformed observed public sector supply share, logit($y_{p,t,m}$), by assuming a Normal distribution likelihood such that,

$$\text{logit}(y_{p,t,m}) \mid \psi_{p,t,m} \sim N(\psi_{p,t,m}, SE^2_{p,t,m}). \tag{12}$$

Where $SE_{p,t,m}$ is associated standard error calculated using the DHS survey microdata and transformed onto the logit scale using the Delta method [40].

## Computation

We used R and JAGS (Just Another Gibbs Sampler) to fit the model. JAGS uses Markov Chain Monte Carlo (MCMC) algorithm and Gibbs Sampling to produce model estimates for Bayesian Hierarchical models [41]. To evaluate the JAGS output we used 'rjags', an R package that offers cross-platform support from JAGS to the R interface [42]. The number of iterations used was 80,000. The burn-in period was set to 10,000. The samples were thinned to every 35$^{th}$ sample. Consequently, the posterior distribution is made up of 2000 samples. To assess convergence, we considered the R-hat and effective sample size (ESS) values of the model parameters using the plot function of rjags, as well as the trace plots and autocorrelation plots of individual parameters [43]. The results were a set of trajectories for the proportion of contraceptive $m$ supplied by the public and private sectors over time for each subnational region included in the study. The mean of these results was taken to be the model's point estimate. The 95% credible intervals were calculated using the 2.5$^{th}$ and 97.5$^{th}$ percentiles from the posterior distribution for each estimate.

## Results

### National and subnational correlations

Fig 2 shows the heat map of the estimated latent correlations between the five contraceptive methods considered in this study informing $\alpha_{p,m}$, the proxy-intercept, at the national (A) and subnational (B) levels across all subnational provinces. At the national level (Fig 2A), the estimated correlations are showing strong positive relationships between the long-term and permanent methods (LAPM). For OC pills, a short-term method, the estimated correlations with the other methods are weakly positive. This trend is in keeping with previous studies that found OC pills are a popular choice for women accessing contraceptive through the private sector [8,10,11]. In contrast, contraceptive methods that require a medical professional for administration are more likely to be accessed through the public sector. These correlations would imply that as the most recently observed public sector supply share of a given LAPM increases, the other LAPMs also tend to increase. For the subnational level correlations (Fig 2B), the estimated correlations between female sterilization and the other methods are the weakest with moderately-strong positive correlations (0.60–0.63). The remaining cross-method correlations are strong between 0.70 (OC pills and implants) and 0.80 (OC pills and IUDs). The estimated cross-method correlations at the subnational level are more homogenous than those observed at the national level. This may be due in part to the noisier signal observed at the subnational level.

### Country case studies

While we produced model estimates based on model fit to the subnational provinces of 24 countries, we have chosen two case-study countries Nigeria and Rwanda to showcase the strength of our modelling approach. Both countries have

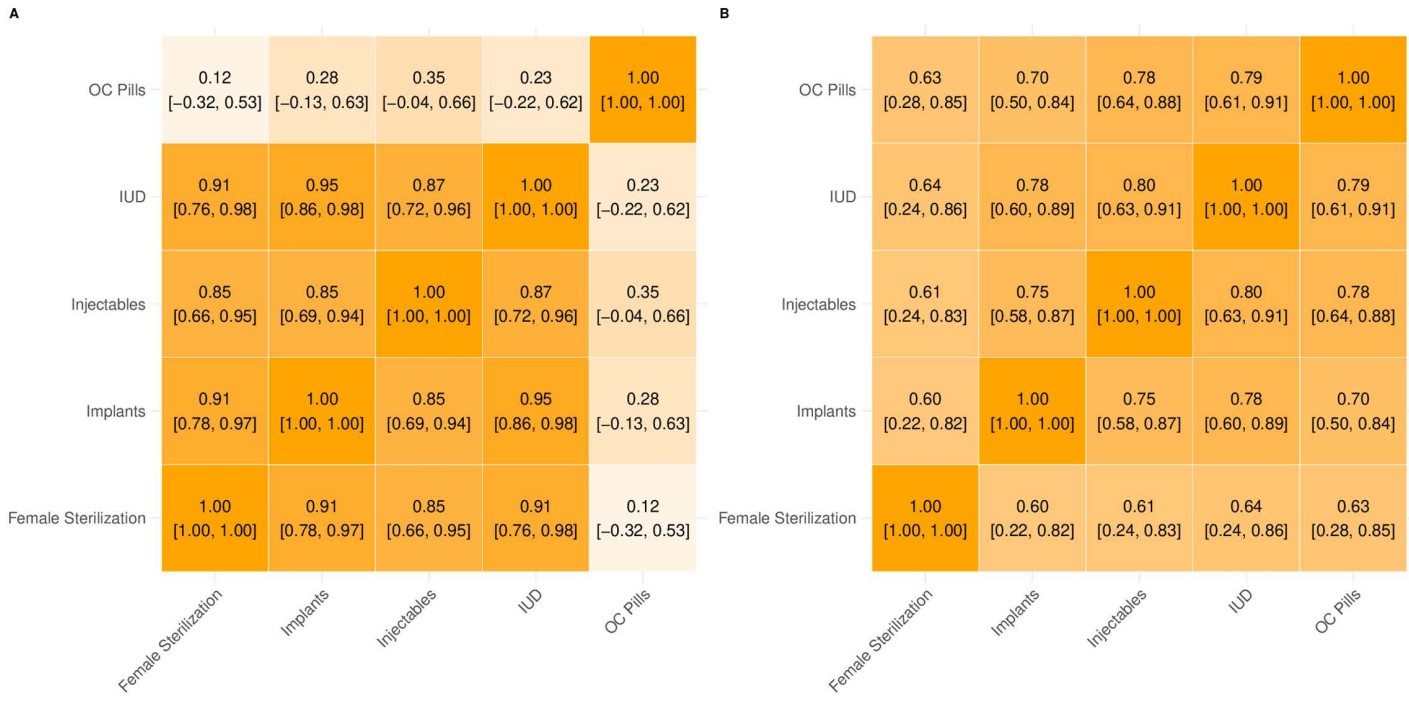

**Fig 2. Estimated (A) national- and (B) subnational-level cross-method correlations informing $\alpha_{p,m}$, the proxy-intercept.** The estimated mean correlations captured by the Wishart prior for the intercept term of the proposed model. The intercept is informed by the most recently observed public supply shares of each method across all (A) countries and (B) subnational administration regions. Each of the five methods is listed along the x- and y-axes, with the estimated mean correlation is given in each square, with the associated 95% credible interval underneath in brackets. The strength of the correlation is emphasized by the depth of the shade. Lighter infer correlations closer to 0, while darker colours infer stronger correlations.

varying amounts of data across the five contraceptive methods and have experienced changes in the modern contraceptive supply share trends over time.

**Nigeria.** Fig 3 shows the estimated proportion of contraceptives supplied by the public and private sectors with uncertainty from 1990 to 2030 across the six geopolitical zones. The boundaries of these zones have been consistent for the duration of the observation period. Historically, only the North Central region of Nigeria has collected data on the supply of female sterilization. As such, the remaining regions are estimated via the hierarchical estimation process on the intercept. The estimated method supply share in the remaining regions is given by the most recently supplied share in the North Central region. With no data to inform any deviation from this observed level, coupled with the splines being penalised to project steadily into the future, we see flat trends occurring for these regions. In the North Central region,

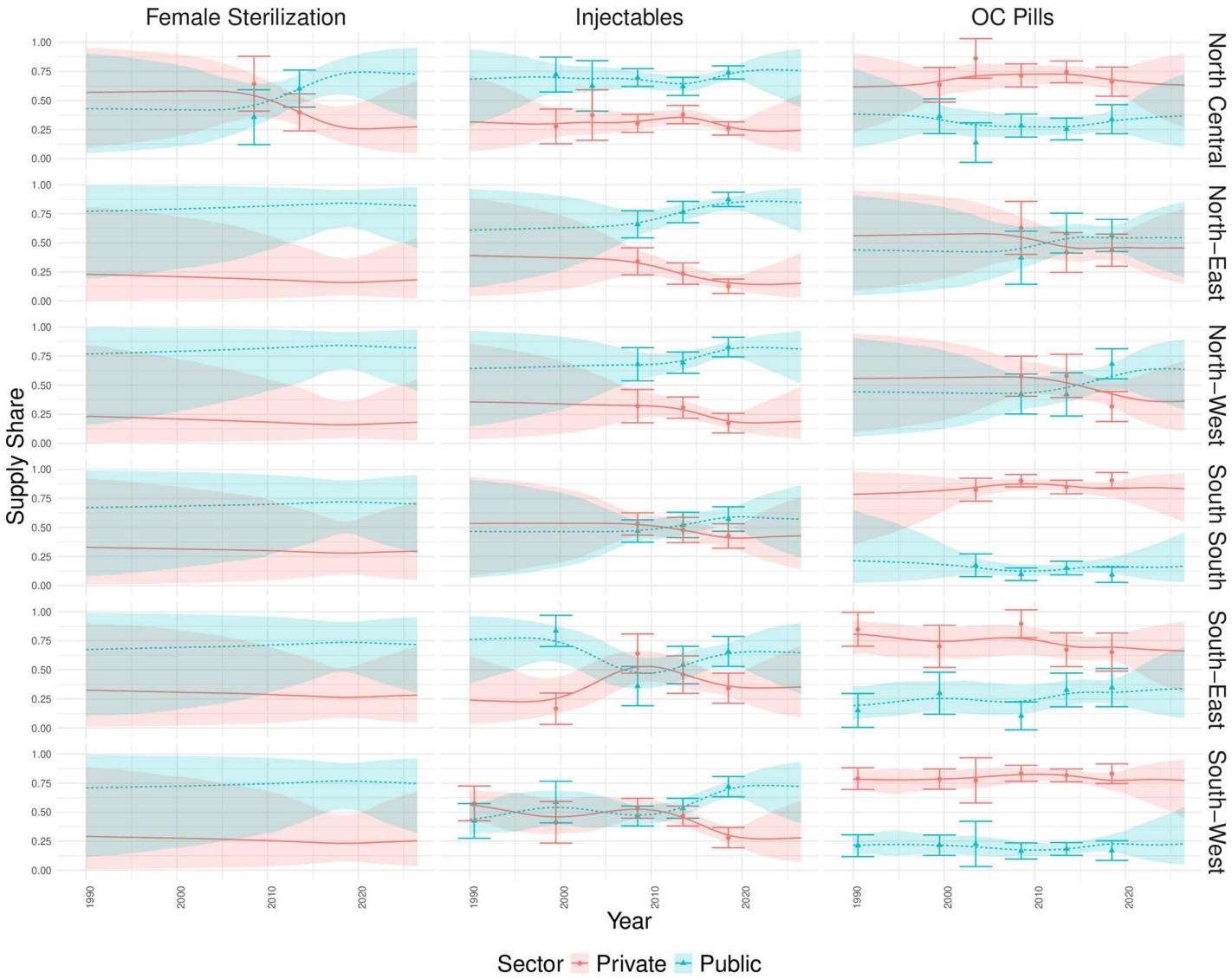

**Fig 3. Estimated subnational contraceptive supply shares in Nigeria with uncertainty.** The projections for the proportion of modern contraceptives supplied by each sector for three of the five contraceptive methods, in the six geopolitical zones of Nigeria. The median estimates are shown by the continuous line while the 95% credible interval is marked by shaded coloured areas. The DHS data point is signified by a point on the graph with error bars displaying the standard error associated with each observation. The sectors are shown by blue triangles for public and red for the private sector.

the model indicates an increasing trend between 2015 and 2020, approximately. In contrast to the data sparsity seen in female sterilization, injectables have between three and five survey observations per region. The flexibility of the splines is demonstrated here as the model is able to capture the complex nature of the data in each region. Historically, OC pills tends mostly to be provided by the private sector across all subnational regions. However, the model estimates indicate that this private sector supply share has been steadily declining over time. This is in contrast to the supply trends observed in other methods, where the public sector is the main supplier across all regions, and has remained relatively constant over time. Some smoothing is taking place as the data model is informed by the observed standard errors of the survey data. For this reason, observations with large associated standard errors are not as closely adhered to, when compared to observations with smaller uncertainty. This is clearly seen in North-Central OC pills where the model estimates smooth through the survey observation at 2003. When we consider the private sector supply shares spatially and over time, we can see geographical trends in the supply of certain methods (Fig 4). For instance with IUDs (Fig 4: IUDs), there appears to be a north-south split across Nigeria in the private sector supply over time. The northern half (North East and North West subnational regions) tend towards very low proportions of IUDs provided by the private sector while the southern half of the country (North Central, South West, South South, South East) sees higher proportions of IUDs supplied by the private sector. This trend is replicated in OC pills (Fig 4: OC Pills) where again, the northern half of the country tends to supply a lower proportion of OC pills via the private sector when compared to the southern half.

**Rwanda.** In Rwanda, we estimate contraceptive method supply shares for the five provinces (Fig 5). The boundaries of these zones have been consistent for the duration of the observation period. Historically, the public sector has dominated the supply of all five contraceptive methods across all five provinces since 1990. The supply of implants is almost exclusively

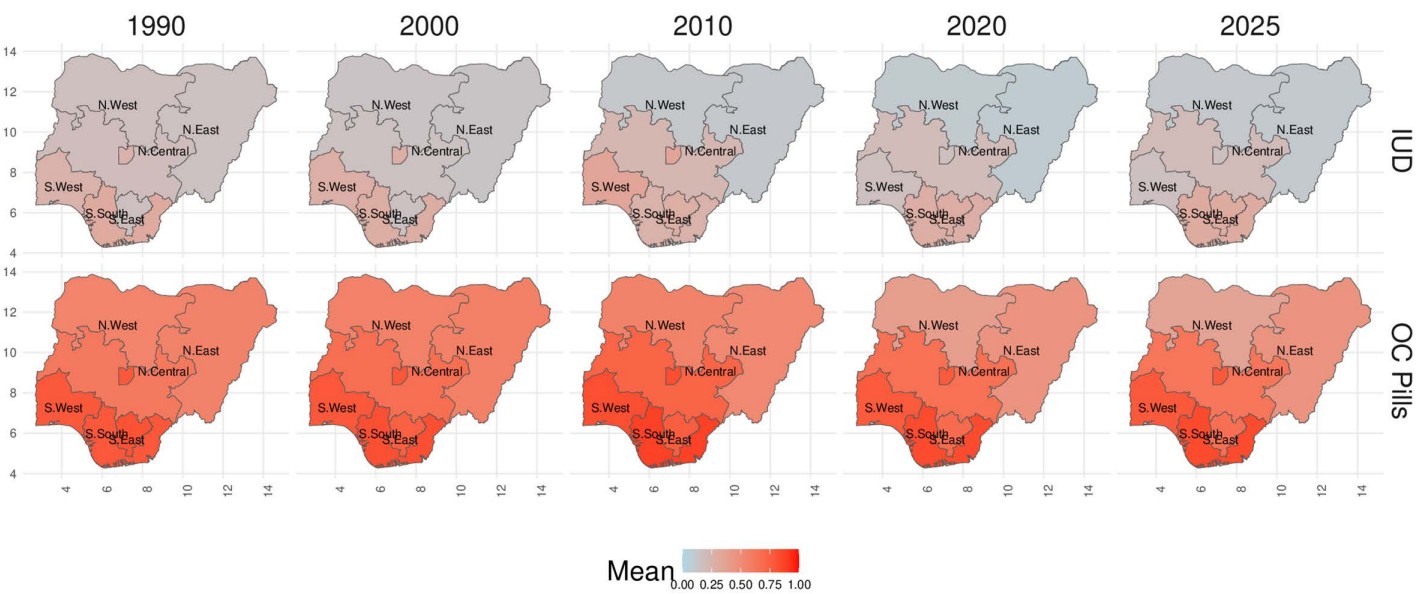

**Fig 4. Map of estimated subnational contraceptive private sector supply shares in Nigeria over time.** The mean estimates of private sector contraceptive method supply shares for two of the five contraceptive methods, in the six geopolitical zones of Nigeria. The colour captures the mean supply shares where light blue colours capture supply shares near 0 while the deep red colour captures supply shares approaching 100%. Five years of estimates are given at 10-yearly intervals from 1990 to 2020 and then the present year, 2025. Label 'IUDs' shows the supply of intra-uterine devices. Label 'OC Pills' shows the supply of oral contraceptive pills. Administrative boundary data were obtained from the IPUMS DHS database (https://www.idhsdata.org/idhs/), derived from The DHS Program spatial data, and are available for research use under a Creative Commons Attribution–compatible license. The figure was created by the authors using ggplot2 in R.

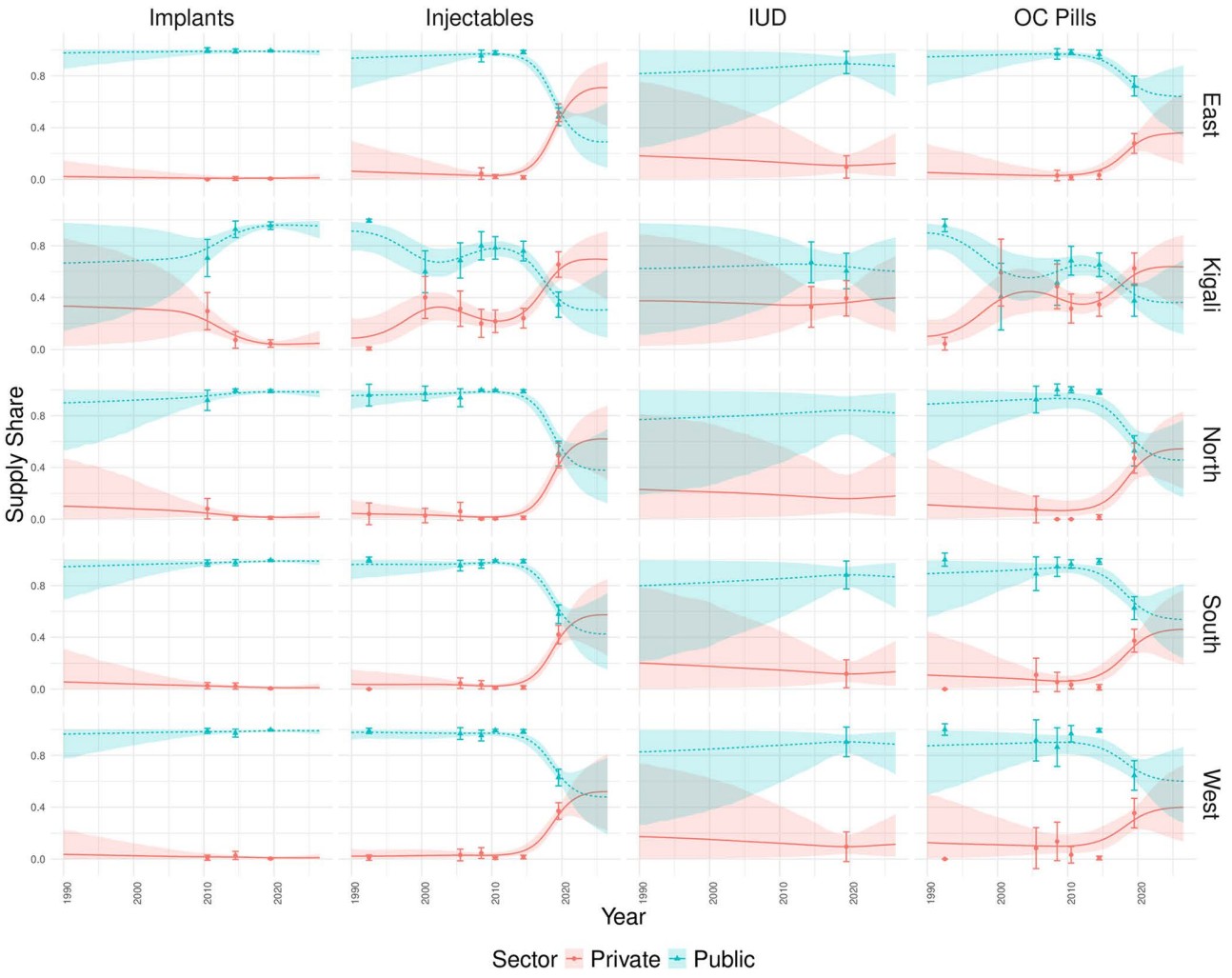

**Fig 5. Estimated subnational contraceptive supply shares in Rwanda with uncertainty.** The projections for the proportion of modern contraceptives supplied by each sector for four of the five contraceptive methods, in the five provinces of Nigeria. The mean estimates are shown by the continuous line while the 95% credible interval is marked by shaded coloured areas. The DHS data point is signified by a point on the graph with error bars displaying the standard error associated with each observation. The sectors are shown by blue triangles for public and red for the private sector.

provided by the public sector, with very small proportions across all methods being privately sourced. The supply of IUDs in Kigali is at an approximately 60:40 split between the public and private sector. This is quite different to the trends we observe in the other regions where the public sector takes over 80% of the market share. The contraceptive supply market has been turbulent in Rwanda across all the regions for injectables and OC pills. In injectables, from 1990 to 2015 the market shares had remained very steady over time with little private sector involvement. However, we see that post-2015 the private sector rapidly increases it's share of the injectables supply market and post-2020 it overtakes the public sector as the largest supplier of injectables in all regions. Similarly, in OC pills we see a rapid increase of private sector supply shares post-2015. In Kagali and the North we see that the private sector is estimated to become the largest supplier of OC pills post-2019. In the South, the OC pill market share is split at an approximate 50:50 between public and private shares. Fig 6 captures the mean trends of the private sector supply shares spatially and temporally. Across all methods, we see that Kigali provinces tends to have the strongest private sector supply share over time. From Fig 6, we see that Kigali tends to increase the private sector supply share

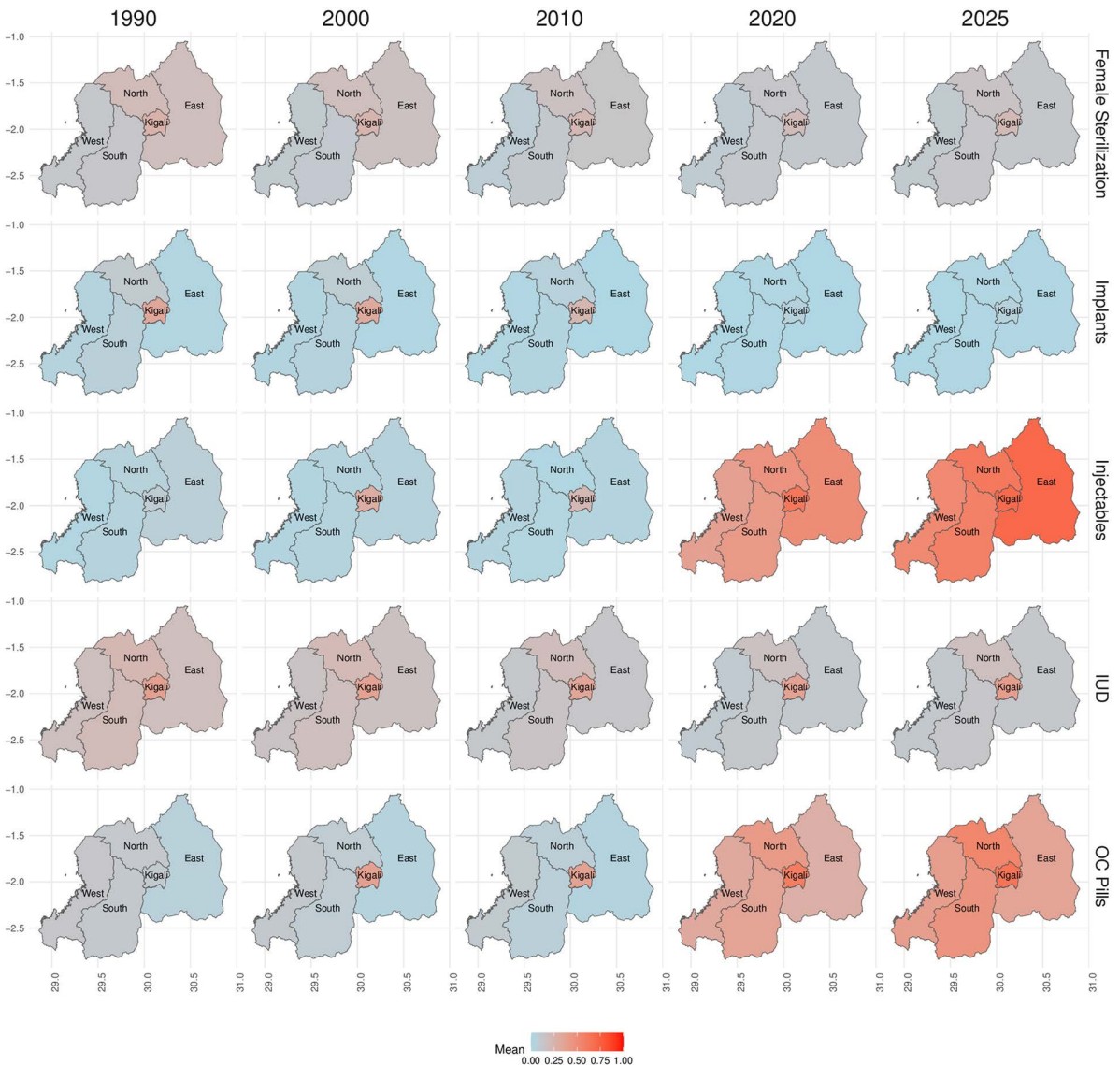

**Fig 6. Map of estimated subnational contraceptive supply shares in Rwanda over time.** The mean estimates of private sector supply shares of the five contraceptive methods across the five provinces of Nigeria. The colour captures the mean supply shares where light blue colours capture supply shares near 0 while the deep red colour captures supply shares approaching 100%. Five years of estimates are given at 10-yearly intervals from 1990 to 2020 and then the present year, 2025. The A label shows the supply of female sterilization. The B label shows the supply of implants. Label C shows the supply of injectables. Label D shows the supply of intra-uterine devices (IUDs). Label E shows the supply of oral contraceptive pills. Administrative boundary data were obtained from the IPUMS DHS database (https://www.idhsdata.org/idhs/), derived from The DHS Program spatial data, and are available for research use under a Creative Commons Attribution–compatible license. The figure was created by the authors using ggplot2 in R.

of a given method and then the neighboring regions follow suit afterwards. Overall, private sector supply shares have historically been low in Rwanda across all regions, with the exception of OC pills (Fig 6E) and injectables (Fig 6C) in recent years.

## 2025 estimate summary

In Table 2, we present the estimated mean method supply shares across all subnational administration regions for the year 2025. Overall, the public sector dominates in the supply of modern contraceptives at the subnational level.

**Table 2. Summary of estimated contraceptive supply shares across all subnational administration regions in 2025 for each method and sector.** The mean estimate for the proportion supplied and the associated standard deviation (SD) are listed.

| | Public | | Private | |
|---|---|---|---|---|
| | Mean | SD | Mean | SD |
| **Female Sterilization** | 0.773 | 0.095 | 0.207 | 0.095 |
| **Implants** | 0.829 | 0.160 | 0.119 | 0.157 |
| **Injectables** | 0.823 | 0.145 | 0.171 | 0.145 |
| **IUD** | 0.768 | 0.150 | 0.234 | 0.150 |
| **OC Pills** | 0.520 | 0.239 | 0.480 | 0.239 |

Implants are most commonly accessed through the public sector with a mean of approximately 83 percentage points. OC pills show an approximate 50:50 split between public and private sector market shares. In comparison to OC pills, a short-term contraceptive method, the long-acting and permanent methods, female sterilization, implants, injectables, and IUDs, have higher public sector supply shares by at least 20 percentage points in each instance. This is consistent with previous research where women were found to be more likely to access short-term methods via private facilities [8,10,11]. These findings are also consistent with the contraceptive method supply shares estimated at the national-level in 2023 [25].

## Validation results

The model described in this paper has been validated using several metrics to assess its effectiveness at estimating subnational method supply shares over time. We split our complete data into testing and training datasets, any data beyond 2015 was labeled as the test set and withheld for validation. Further details of the validation approach and the metrics used to evaluate it's performance can be found in the Supplementary Materials. Given the complex nature of the data, the model is performing reasonably well. Table 3 gives the validation results for the Multivariate intercept P-spline model, described in the Methods section, and the modelling alternatives, described in the Supplementary Materials. First we consider the mean absolute relative error (MARE), a negatively orientated accuracy measure of each approach using the test set. The Multivariate intercept P-spline model has the lowest MARE at approximately 7 percentage points while the 0-covariance P-spline model saw the largest MARE at approximately 12 percentage points. We also consider the standardized absolute prediction error (SAPE), which is a measure of the dispersion of the generated predictive distributions for the test set. In all modelling cases, the SAPE was greater than 1. This would imply that the data is more spread out that the models are predicting. The lowest SAPE was recorded with the Fully Multivariate P-spline model at 2.81 while the largest SAPE was recorded for the shrinkage P-spline model. Our proposed model had the second smallest SAPE at 3.46. For coverage, in the 80% instance we would expect our models to capture 80% of the test set. There was some over-fitting for this metric, with the Multivariate intercept P-spline model giving a coverage of approximately 82%. This was the closest coverage to the expected 80% of all 5 models. The shrinkage P-spline model had the lowest coverage at approximately 77%. In the 95% case, we expect our models to capture 95% of the test set. In this instance, we do not see the same over-fitting issue as the 80% coverage metric. The fully multivariate P-spline model scores slightly higher than the proposed model in this instance with an approximate coverage of 95%. However, the proposed model has a marginally smaller coverage of approximately 94 percentage points. To evaluate the accuracy of our model, we considered the root mean square error (RMSE) of the public sector test set estimates. The lowest RMSE is observed in our proposed model with an approximately average error of 15 percentage points. The highest RMSE was observed in the Multivariate delta P-spline model at approximately 19 percentage points. The width of the prediction intervals (PI width) is a crucial validation metric because it directly impacts the usability of the estimates. If the PI width is too narrow, the intervals may fail to capture the true values, leading to poor coverage. On the other hand, if the PI width is too wide, the estimates

**Table 3. Model validation results.** The validation results for the test set across different potential modelling approaches. MARE is mean absolute relative error and is a percentage. SAPE is the standardized absolute prediction error and is measured on the logit scale. Coverage is the proportion of the test set observations that are captured within the 80% and 95% prediction intervals produced by the model. RMSE is root mean square error. We consider the median PI width and evaluate the location of the incorrectly estimated leave-one-out validation test set observations. The Multivariate intercept P-spline model is described in the Methods section, while the modelling alternatives are described in the Supplementary Materials.

| Metric | | Multivariate intercept P-spline model | Multivariate delta P-spline model | 0-covariance P-spline model | Shrinkage P-spline model | Fully multivariate P-spline model |
|---|---|---|---|---|---|---|
| MARE (%) | | 7.42 | 11.25 | 11.86 | 11.16 | 9.10 |
| SAPE | | 3.46 | 2.94 | 3.55 | 3.80 | 2.81 |
| 80% coverage (%) | | 81.95 | 82.54 | 77.51 | 76.62 | 82.54 |
| 95% coverage (%) | | 93.19 | 92.01 | 91.42 | 90.82 | 93.49 |
| RMSE (%) | | 15.26 | 19.39 | 17.97 | 18.85 | 16.19 |
| Median 95% PI width (%) | | 46.50 | 52.60 | 47.40 | 44.1 | 50.9 |
| Location of incorrectly estimated observations | *Above the 95% PI* | 2.37 | 2.07 | 3.55 | 3.55 | 1.48 |
| | *Below the 95% PI* | 4.44 | 5.92 | 4.73 | 5.62 | 5.03 |

become less informative and lose their practical value in other modelling scenarios. The smallest median 95% PI width was observed in the shrinkage P-spline model at approximately 44 percentage points. The largest PI width was observed for the Multivariate delta P-spline model at approximately 53 percentage points. Notably, the Multivariate intercept P-spline model provides the second smallest PI width at approximately 47 percentage points. This is 5 percentage points lower than the Multivariate delta P-spline model. Finally, in an unbiased model we would expect to see an even split of the incorrectly estimated test set observations above and below the prediction interval bounds. In all modelling instances, there are a higher proportion of incorrectly estimated observations below the 95% prediction interval. This would imply that the models tend to over-estimate the data. Given these results, we conclude that the proposed Multivariate intercept P-spline model is the performs the best overall. It has good coverage in both the 80% and 95% instances. The width of the prediction interval is not as large as other modelling alternatives, indicating that this updated model has lower uncertainty in the model estimates compared to the others. In addition, it has the lowest RMSE and MARE, and the second lowest SAPE value of all the models. Overall, the method seems to have provided reasonably accurate and well-calibrated probabilistic projections for the 2015–2022 period.

## Discussion

In this paper, we have developed a Bayesian method for probabilistic projections of subnational contraceptive supply shares over time with available DHS survey data. The modelling framework is based on Bayesian hierarchical models and uses penalized splines to capture the evolution of the contraceptive method supply from the public and private sectors while imposing a correlation structure to capture correlations between the most recently observed public sector supply shares. The hierarchical nature of the modelling framework allows for cross-subnational information sharing within countries to promote precise estimation, even when limited data is available. The posterior predictive distributions are estimated using past DHS data for all subnational administration regions from 1990 to 2022. The model will produce estimates within the period of available survey data as well as projections beyond the most recent data point. The resulting predictive distributions were accurate and reasonably well calibrated in an out of sample validation exercise for forecasting the most recent seven year period. The modelling framework was found to outperform other suitable alternative modelling approaches.

This modelling framework has several advantages over the aforementioned modelling alternatives, and makes another contribution to the estimation of subnational proportions. Firstly, using the Bayesian hierarchical framework accounts for

the spatial nature of the data while also allowing subnational regions within a country to share information. An advantage of this is that information sharing within countries improves the precision of the resulting subnational estimates and informs model estimates where previously no data was present, all in a data-driven approach. Secondly, using a multivariate approach to hierarchal modelling accounts for the correlations that exist between method supply shares, and allows for another information exchange to occur, this time across contraceptive methods. Estimating these cross-method national- and subnational-level correlations further explains the latent trends underpinning the complexity nature of subnational method supply shares. Incorporating the observed standard errors of the DHS estimates into the data model further controls the uncertainty associated with the resulting estimates, and promotes the smooth model estimates and projections from an otherwise heterogeneous dataset. Finally, using penalised splines allows for data driven, flexible model-based estimates. For these reasons, we believe that proposed model makes a valuable contribution to the area of subnational estimation of proportions, in data sparse settings.

In addition, the approach introduces two key improvements over previous approaches of Comiskey et al. (2023) and Comiskey et al. (2024). First, it captures non-zero covariances between proxy-intercepts rather than between rates of change. Since subnational data tends to be noisier than national-level supply share data, correlations between rates of change are often difficult to estimate and uncertain. By allowing the most recently observed levels to share information across methods, this approach enhances estimate precision in data-sparse regions. Second, the model adopts a fully Bayesian process for estimating covariance matrices, replacing the earlier method of Comiskey et al. (2024) that relied on *a posteriori* correlation estimates. This update is not only more computationally efficient but also avoids restricting the parameter space of the resulting covariance matrix. Overall, this updated model more effectively captures the complexities of subnational data, while still offering insights into cross-method correlations and spatial relationships found in the previous approach of Comiskey et al. (2024).

In recent years, a substantial number of low- and middle-income countries have implemented a dencentralised health sector making the estimation of health indicators in data-sparse, small-areas of vital importance to researchers and policymakers [12]. However, the lack of reliable data and regular subnational estimates of key indicators have reduced the ability of health and development authorities' to strengthen the delivery of contraception and other reproductive health services via local systems [14]. When evaluating health programmes, policymakers must consider cost, access and quality [44]. A potential application of these subnational method supply shares is in the evaluation of access. Strengthening access to contraceptives through public-private sector collaborations and complementing ongoing public-private partnerships for the advocacy of family planning can lead to an increase in contraceptive uptake [45]. It is well-established that teenagers and sexually active unmarried women tend to use private sector suppliers of contraception [46]. However, as seen in our country case studies of subnational contraceptive supplied in Nigeria and Rwanda, many contraceptives are supplied almost solely by the public sector across many subnational regions. These model estimates provide valuable insights for policymakers into the temporal and spatial dynamics of the subnational contraceptive supply share markets, highlighting areas for potential improvement in supply dynamics and equity. Having public and private sectors both supplying contraceptives is key to achieving equity among family planning users [11], but our case studies have shown that for some contraceptives the private sector is supplying very low proportions of some contraceptives over time. Lastly, a total-market approach is essential to the success and sustainability of the family planning market. Strengthening supply chains through informed data-driven supply share estimates will contribute to ensuring contraceptive security for women and girls globally [47].

In future work, we hope to incorporate additional demographic and family planning indicator covariates into the modelling framework to further improve the precision of the model estimates. We hope to extend the framework to account for multiple data sources, accounting for the different sources of error.

## Supporting information

**S1 File. Supplementary Materials: Bayesian probabilistic projections of proportions with limited data [48–52].** (PDF)

## Author contributions

**Conceptualization:** Hannah Comiskey, Niamh Cahill, David T. Fraizer, Worapree Maneesoonthorn, Leontine Alkema.

**Data curation:** Hannah Comiskey.

**Formal analysis:** Hannah Comiskey.

**Methodology:** Hannah Comiskey, Niamh Cahill, David T. Fraizer, Worapree Maneesoonthorn.

**Software:** Hannah Comiskey.

**Supervision:** David T. Fraizer, Worapree Maneesoonthorn.

**Validation:** Hannah Comiskey.

**Visualization:** Hannah Comiskey.

**Writing – original draft:** Hannah Comiskey.

**Writing – review & editing:** Niamh Cahill, David T. Fraizer, Worapree Maneesoonthorn, Leontine Alkema.

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
