## [Decision Letter · Decision Letter 0]

7 Nov 2025

Dear Dr. Comiskey,

Thank you for submitting your manuscript to PLOS ONE. After careful consideration, we feel that it has merit but does not fully meet PLOS ONE’s publication criteria as it currently stands. Therefore, we invite you to submit a revised version of the manuscript that addresses the points raised during the review process.

We look forward to receiving your revised manuscript.

Kind regards,

Juan Pablo Gutierrez

Academic Editor

PLOS ONE

**Journal Requirements:**

“Worapree Maneesoonthorn and Hannah Comiskey gratefully acknowledge support by the Australian Research Council through grant DP200101414. David Frazier gratefully acknowledges funding from the Australian Research Council under Projects DE200101070 and DP200101414.”

3. Please note that your Data Availability Statement is currently missing the DOI/accession number of each dataset. If your manuscript is accepted for publication, you will be asked to provide these details on a very short timeline. We therefore suggest that you provide this information now, though we will not hold up the peer review process if you are unable.

4. Please ensure that you refer to Figure 5 in your text as, if accepted, production will need this reference to link the reader to the figure.

5. We note that Figures 4 and 6 in your submission contain mapimages which may be copyrighted. All PLOS content is published under the Creative Commons Attribution License (CC BY 4.0), which means that the manuscript, images, and Supporting Information files will be freely available online, and any third party is permitted to access, download, copy, distribute, and use these materials in any way, even commercially, with proper attribution. For these reasons, we cannot publish previously copyrighted maps or satellite images created using proprietary data, such as Google software (Google Maps, Street View, and Earth). For more information, see our copyright guidelines: http://journals.plos.org/plosone/s/licenses-and-copyright.

a. You may seek permission from the original copyright holder of Figure(s) [#] to publish the content specifically under the CC BY 4.0 license.

6. Please upload a copy of Supporting Information figure/table which you refer to in your text on page 10.

**Additional Editor Comments:**

Please review the comments from the reviewers and address them accordingly. Provide the additional details requested to assess the methodology.

Reviewers' comments:

Reviewer's Responses to Questions

**Comments to the Author**

1. Is the manuscript technically sound, and do the data support the conclusions?

Reviewer #1: Yes

Reviewer #2: Yes

2. Has the statistical analysis been performed appropriately and rigorously?

Reviewer #1: Yes

Reviewer #2: Yes

3. Have the authors made all data underlying the findings in their manuscript fully available?

Reviewer #1: Yes

Reviewer #2: No

4. Is the manuscript presented in an intelligible fashion and written in standard English?

Reviewer #1: Yes

Reviewer #2: Yes

Reviewer #1: The paper proposes a novel approach to produce Bayesian probabilistic projections of contraceptive method supply shares at the subnational level, especially in data-sparse settings. The authors state that their modelling framework builds on Bayesian hierarchical models and uses penalized splines, leveraging the spatial nature of data and incorporating a correlation structure between recent supply share observations at national and subnational levels. They explicitly mention improvements over previous approaches, including those by Comiskey et al., 2024, by capturing non-zero covariances between proxy-intercepts and adopting a fully Bayesian process for estimating covariance matrices. This demonstrates original research by extending existing methodologies to address a specific, complex problem in a new way.

Here are the few points to address which can strengthen the paper:

In Introduction section:

• Novelty vs. prior work: Make the distinction from Comiskey et al. (2024) (or any prior national-level model) explicit in the Introduction (authors do make distinctions from Comiskey et al. (2024) and other prior approaches, but primarily in the "Methods" and "Discussion" sections, rather than explicitly in the Introduction).

• Scope and generalisability: Clarify to what extent the framework can be ported to other small-area proportion estimation problems (e.g., immunisation source, ANC service mix). A brief paragraph on external applicability will broaden impact.

In Data section:

• Briefly justify the choice of FP2030 countries and administrative unit definitions. It would be good to Include a short table summarising: country, admin level used, number of regions, survey years.

• It would be good to state why penalised splines were preferred (e.g., smoother long-term trends, fewer parameters) and whether a RW1/RW2 (Random Walks) alternative was tested.

• It would be interesting to report the posterior summaries of the off-diagonal covariance terms (e.g., specific numerical values for the correlations or their magnitudes) to demonstrate their substantive nature.

• The paper will strengthen with explicitly showing sensitivity analysis (e.g., by testing LKJ priors or a separation strategy) to demonstrate robustness.

• While the paper clearly explains its use of a normal-on-logit likelihood, it does not explicitly discuss the implications of this choice when proportions are at or near 0 or 1, where the logit transformation can lead to infinite values and large variances.

• The paper does not provide any justification for the sufficiency of the number of draws (4000 total) based on Effective Sample Size (ESS) metrics, also it does not report any ESS statistics for key parameters. Reporting ESS would indeed be valuable for demonstrating the robustness and efficiency of the sampling process.

Lastly, the paper uses a mix of British and American language. Please choose one and make it coherent throughout the paper. For example in some places the author have used localized and in some localised, similarly modelling and modelling, decentralized and desentralised.

Reviewer #2: This manuscript proposes a method for projecting the proportions of each

contraceptive method at the subnational level. This is an important

problem that has already been addressed at the national level by

Comiskey et al (2024). The manuscript adapts the method to subnational

regions. It also extends the method by accounting for cross-method

correlations, which was not previously done.

It also simplifies the previous method by replacing the previous three-level

model by a two-level model. Academic publication tends to reward

ever-increasing complication, and this goes in the opposite direction.

It has several advantages, including greater simplicity of implementation, ease

of interpretation, and ease of communication to users.

So this simplification is actually an advance, and is a strong point of

the manuscript.

My main concern has to do with the spatial modeling.

The claim is made that spatial correlation is included in the modelling,

but I couldn't find how that is done described in the manuscript.

I would have expected to find it in the process model or data model section.

This makes it hard to evaluate the modelling approach overall.

.

Reviewer #1: **Yes:** Adnan KhanAdnan KhanAdnan KhanAdnan Khan

Reviewer #2: No

---

## [Author Response · Author response to Decision Letter 1]

1 Feb 2026

We would like to thank the editor and the reviewers for their helpful comments and feedback. These constructive comments have improved the paper overall. We have revised the manuscript in response to the comments. The main changes to the paper are;

1. Addition of a cross-country level average to have a complete hierarchical set up for the intercepts. This update did not affect parameter estimates or validation, however for completeness it is import to include it.

2. Updated the introduction section to highlight the potential of the modelling approach to other small-area estimation problems.

3. Updated explanations of the modelling approach, including a more detailed explanation in the context of spatial analysis.

4. Added an explanation of the data imputation approach to the Supplementary Materials.

In what follows, points and comments from the reviewers are highlighted in red and our responses are in black. The line numbers for the updated sections are listed in the response. In the updated PDF document, the below cited passages of text are highlighted in blue.

Journal Requirements:

We have updated the file names to meet the PLOS ONE's style requirements.

“Worapree Maneesoonthorn and Hannah Comiskey gratefully acknowledge support by the Australian Research Council through grant DP200101414. David Frazier gratefully acknowledges funding from the Australian Research Council under Projects DE200101070 and DP200101414.”

We have updated the Supplementary materials to include this information (lines 623-627):

“Worapree Maneesoonthorn and Hannah Comiskey gratefully acknowledge support by the Australian Research Council through grant DP200101414. David Frazier gratefully acknowledges funding from the Australian Research Council under Projects DE200101070 and DP200101414. The funders had no role in study design, data collection and analysis, decision to publish, or preparation of the manuscript.”

3. Please note that your Data Availability Statement is currently missing the DOI/accession number of each dataset. If your manuscript is accepted for publication, you will be asked to provide these details on a very short timeline. We therefore suggest that you provide this information now, though we will not hold up the peer review process if you are unable.

We have updated the Supplementary materials to include this information (lines 480-483):

“The complete set of subnational supply share estimates for every country and subnational administration region can be reproduced using the code and data found on the github, https://github.com/hannahcomiskey/Comiskey_PlosOnepaper.”

4. Please ensure that you refer to Figure 5 in your text as, if accepted, production will need this reference to link the reader to the figure.

We have addressed this typo. Line 308 now reads,

“In Rwanda, we estimate contraceptive method supply shares for the five provinces (Fig. 5).”

5. We note that Figures 4 and 6 in your submission contain mapimages which may be copyrighted. All PLOS content is published under the Creative Commons Attribution License (CC BY 4.0), which means that the manuscript, images, and Supporting Information files will be freely available online, and any third party is permitted to access, download, copy, distribute, and use these materials in any way, even commercially, with proper attribution. For these reasons, we cannot publish previously copyrighted maps or satellite images created using proprietary data, such as Google software (Google Maps, Street View, and Earth). For more information, see our copyright guidelines: http://journals.plos.org/plosone/s/licenses-and-copyright.

These maps are generated using the R package ggplot2 and freely available shape files from IPUMS.

6. Please upload a copy of Supporting Information figure/table which you refer to in your text on page 10.

We have addressed this typo, it now reads:

“Further details of the validation approach and the metrics used to evaluate it's performance can be found in the Supplementary Materials.”

Reviewer #1:

The paper proposes a novel approach to produce Bayesian probabilistic projections of contraceptive method supply shares at the subnational level, especially in data-sparse settings. The authors state that their modelling framework builds on Bayesian hierarchical models and uses penalized splines, leveraging the spatial nature of data and incorporating a correlation structure between recent supply share observations at national and subnational levels. They explicitly mention improvements over previous approaches, including those by Comiskey et al., 2024, by capturing non-zero covariances between proxy-intercepts and adopting a fully Bayesian process for estimating covariance matrices. This demonstrates original research by extending existing methodologies to address a specific, complex problem in a new way.

In Introduction section:

• Novelty vs. prior work: Make the distinction from Comiskey et al. (2024) (or any prior national-level model) explicit in the Introduction (authors do make distinctions from Comiskey et al. (2024) and other prior approaches, but primarily in the "Methods" and "Discussion" sections, rather than explicitly in the Introduction).

An extensive explanation for the distinction from Comiskey et al. 2024 is now given between lines 53 and 74:

“Previous works modelled the distribution of contraceptive method supply shares at the national level using a Bayesian hierarchical penalized spline model, and then explored the application of this modelling framework to subnational data (Comiskey et al. 2024, Comiskey et al. 2023). While performing relatively well across a range of validation measures, this paper revisits the modelling framework of Comiskey et al. (2024) with the aim of improving its suitability for estimating and projecting proportions, while addressing the complex structure of subnational DHS administrative data. In this article, we describe a Bayesian hierarchical penalized spline model that produces annual, subnational, and method-specific estimates of the proportion of modern contraceptives coming from the public sector. In the context of SAE, Bayesian hierarchical models allow for the pooling of information across larger populations to inform smaller sub-population estimates, where data is sparser. As a result, these models provide a powerful framework for estimating subnational method supply shares. The Comiskey et al. (2024) approach also estimates and incorporates cross-method correlations to inform the rates of change in spline coefficients. However, due to the heterogeneity of subnational data, capturing cross-method correlations within the rates of change between spline coefficients becomes difficult. Instead, we present an approach that captures the cross-method correlations for the most recently observed survey levels of public sector supply shares. Finally, Comiskey et al. (2024) estimate a multivariate compositional outcome across three-way breakdown of the contraceptive supply market. Due to small sample sizes at the subnational level, we instead focus on a simplified public–private breakdown of the outcome.”

• Scope and generalisability: Clarify to what extent the framework can be ported to other small-area proportion estimation problems (e.g., immunisation source, ANC service mix). A brief paragraph on external applicability will broaden impact.

We have added in a paragraph (lines 87-95) discussing the generalisability of this method:

“While our focus is the specific supply-share monitoring scenario, we note that this more parsimonious modeling approach is highly generalisable and can be applied to a wide range of datasets beyond the current context. By structuring the model to capture both hierarchical dependencies and flexible distributional assumptions, it can accommodate different levels of aggregation, varying sample sizes, and diverse outcome types; e.g., the proportion of women who received antenatal care from a skilled provider or proportion of children vaccinated between 12-23 months”

In Data section:

• Briefly justify the choice of FP2030 countries and administrative unit definitions. It would be good to Include a short table summarising: country, admin level used, number of regions, survey years.

We have updated the data source section (lines 127-131) to include the following justification:

“The countries included in this study are selected from those participating in the FP2030 initiative with IPUMS-DHS data available after 2012 and were also previously considered in Comiskey et al. 2024. There are 23 countries containing 160 subnational regions, included in this study. The administration level used was the integrated geographic units calculated by IPUMS. Using the integrated geographic units ensured that the data had consistent boundaries across all years.”

• It would be good to state why penalised splines were preferred (e.g., smoother long-term trends, fewer parameters) and whether a RW1/RW2 (Random Walks) alternative was tested.

We are currently employing an RW1 approach to the estimation of the spline coefficients. Higher order RW can be recovered if supported by the data. An explanation for the use of splines is added on lines 173-176:

“The use of a random walk model of order 1 to estimate the coefficients of the penalised splines allows us to flexibly model non-linear trends, while the penalty term controls the degree of smoothness and prevents overfitting.”

• It would be interesting to report the posterior summaries of the off-diagonal covariance terms (e.g., specific numerical values for the correlations or their magnitudes) to demonstrate their substantive nature.

Figure 2 (line 290) shows a heat map of the estimated mean correlations with associated 95% credible intervals captured by the Wishart prior for the intercept term of the proposed model. Estimated (A) national- and (B) subnational-level cross-method correlations informing \alpha_{p,m}, the proxy-intercept.

• The paper will strengthen with explicitly showing sensitivity analysis (e.g., by testing LKJ priors or a separation strategy) to demonstrate robustness.

We carried out a sensitivity analysis using a more conservative Wishart prior. We did not use an LKJ prior as it is not available in JAGS. Furthermore, we found that a separation strategy led to model estimates that did not converge. Using the more conservative priors, we found the correlations remained unchanged. We updated the Supplementary Materials to include this detail (lines 613-621):

‘To assess the robustness of our estimated correlations, we conducted a prior sensitivity analysis by increasing the degrees of freedom in the Inverse-Wishart prior used for the variance–covariance matrices of the spline coefficients. Specifically, we considered a more informative prior of the form

\mathrm{\Sigma}_\theta ∼ IW (I_M , M + 5), \mathrm{\Sigma}_\alpha ∼ IW (I_M, M + 5),

where the additional degrees of freedom impose stronger shrinkage toward the identity matrix. This in turn induces greater shrinkage of the off-diagonal elements, pulling the implied correlations closer to zero. We found no significant difference in the estimated correlations from this more conservative prior.’

Below is a side-by-side comparison of the correlations estimated with M+1 degrees of freedom and M+5 degrees of freedom:

Figure 1. Estimated (A) national- and (B) subnational-level cross-method correlations informing \alpha_{p,m}, the proxy-intercept estimated from an Inverse Wishart prior with M+1 degrees of freedom. The estimated mean correlations captured by the Wishart prior for the intercept term of the proposed model. The intercept is informed by the most recently observed public supply shares of each method across all (A) countries and (B) subnational administration regions. Each of the five methods is listed along the x- and y-axes, with the estimated mean correlation is given in each square. The strength of the correlation is emphasised by the depth of the shade. Lighter infer correlations closer to 0, while darker colours infer stronger correlations.

Figure 2. Estimated (A) national- and (B) subnational-level cross-method correlations informing \alpha_{p,m}, the proxy-intercept estimated from an Inverse Wishart prior with M+5 degrees of freedom. The estimated mean correlations captured by the Wishart prior for the intercept term of the proposed model. The intercept is informed by the most recently observed public supply shares of each method across all (A) countries and (B) subnational administration regions. Each of the five methods is listed along the x- and y-axes, with the estimated mean correlation is given in each square. The strength of the correlation is emphasised by the depth of the shade. Lighter infer correlations closer to 0, while darker colours infer stronger correlations.

• While the paper clearly explains its use of a normal-on-logit likelihood, it does not explicitly discuss the implications of this choice when proportions are at or near 0 or 1, where the logit transformation can lead to infinite values and large variances.

A paragraph has been added to the Data sources section (lines 133-142) have been added that explain how we deal with exact 0s and 1s:

“To avoid issues with exact zeros or ones, on the logit scale, the data was condensed using the `lemon-squeezer' method. This simple transformation slightly shrinks all data points away from 0 and 1 while maintaining their relative ordering and approximate scale. To address issues with small sample size, we filtered the database to only include observations where at least one sector (public or private) has a sample size of at least 10 women. This removes sets of observations with large uncertainty due to small sample sizes. Sampling errors were calculated while accounting for the sampling design using a Taylor series linearisation method to approximate the standard error of the calculated proportions. A complete description of this imputation process is described in the Supplementary Materials”

• The paper does not provide any justification for the sufficiency of the number of draws (4000 total) based on Effective Sample Size (ESS) metrics, also it does not report any ESS statistics for key parameters. Reporting ESS would indeed be valuable for demonstrating the robustness and efficiency of the sampling process.

On line 240-243, we have clarified the description of our convergence assessment. We use of R-hat values to evaluate convergence and ESS to evaluate autocorrelation. For parameters with lower ESS, we considered their individual autocorrelation plots and deemed them to be sufficient. The publicly available R vignette to reproduce these results includes the functions to look at these diagnostics so that users can produce the plots themselves.

“To assess convergence, we considered the R-hat and effective sample size (ESS) values of the model parameters using the plot function of rjags, as well as the trace plots and autocorrelation function plots of individual parameters”

• Lastly, the paper uses a mix of British and American language. Please choose one and make it coherent throughout the paper. For example in some places the author have used localized and in some localised, similarly modelling and modelling, decentralized and decentralised.

The spelling has been adjusted for consistency.

Reviewer #2:

This manuscript proposes a method for projecting the proportions of each contraceptive method at the subnational level. This is an important problem that has already been addressed at the national level by Comiskey et al (2024). The manuscript adapts the method to subnational regions. It also extends the method by accounting for cross-method correlations, which was not previously done. It also simplifies the previous method by rep

---

## [Decision Letter · Decision Letter 1]

6 Mar 2026

Bayesian probabilistic projections of proportions with limited data: An application to subnational contraceptive method supply shares.

PONE-D-25-25522R1

Dear Dr. Comiskey,

We’re pleased to inform you that your manuscript has been judged scientifically suitable for publication and will be formally accepted for publication once it meets all outstanding technical requirements.

Kind regards,

Juan Pablo Gutierrez

Academic Editor

PLOS One

Additional Editor Comments (optional):

Reviewers' comments:

Reviewer's Responses to Questions

**Comments to the Author**

Reviewer #2: All comments have been addressed

2. Is the manuscript technically sound, and do the data support the conclusions?

Reviewer #2: Yes

3. Has the statistical analysis been performed appropriately and rigorously?

Reviewer #2: Yes

4. Have the authors made all data underlying the findings in their manuscript fully available?

Reviewer #2: Yes

5. Is the manuscript presented in an intelligible fashion and written in standard English?

Reviewer #2: Yes

Reviewer #2: (No Response)

.

Reviewer #2: No

---

## [Editor Report · Acceptance letter]

PONE-D-25-25522R1

PLOS One

Dear Dr. Comiskey,

I'm pleased to inform you that your manuscript has been deemed suitable for publication in PLOS One. Congratulations! Your manuscript is now being handed over to our production team.

Kind regards,

on behalf of

Dr. Juan Pablo Gutierrez

Academic Editor

PLOS One